# Combination of Expanded Allogeneic NK Cells and T Cell-Based Immunotherapy Exert Enhanced Antitumor Effects

**DOI:** 10.3390/cancers15010251

**Published:** 2022-12-30

**Authors:** Xiao Wang, Xuejiao Yang, Yueping Wang, Yunshuo Chen, Ying Yang, Siqi Shang, Wenbo Wang, Yueying Wang

**Affiliations:** 1Shanghai Institute of Hematology, State Key Laboratory of Medical Genomics, National Research Center for Translational Medicine at Shanghai, Ruijin Hospital, Shanghai Jiao Tong University School of Medicine, Shanghai 200025, China; 2Institute of Interdisciplinary Sciences, Shanghai University of Traditional Chinese Medicine, Shanghai 201203, China; 3Department of Oncology, Shanghai Tenth People’s Hospital, Tongji University School of Medicine, Shanghai 200072, China; 4LeaLing Biopharma Company, Ltd., Suzhou 215000, China

**Keywords:** natural killer cells, tumor-reactive T cells, T cell receptor-engineered T cells, human leukocyte antigen, combination immunotherapy

## Abstract

**Simple Summary:**

The clinical efficacy of neoantigen-reactive tumor-infiltrating lymphocytes and T cell receptor-engineered T cells (TCR-T) therapy is primarily attributed to the effective CD8^+^ T cell responses that are limited to HLA-I presenting tumor-specific antigens. NK cells have the therapeutic potential for refractory tumors that develop immune evasion due to HLA-I downregulation. In this study, we compare different culture systems and solve the challenge of the large-scale expansion of NK cells with strong cytotoxic activity. We attempt to seek the transcriptional alteration of NK cells expanded by different culture conditions and identify the genomic profiles of optimized NK cells with strong cytotoxicity. Here, we demonstrate that the cooperation of “off-the-shelf” NK cells with tumor-activated alloreactive T cells or with NY-ESO-1-specific 1G4 TCR-T cells further enhanced tumors lysis, especially against tumors with HLA-I downregulation. Our study improves the immune incompetence caused by HLA-I downregulation and innovates future combination strategies for cancer immunotherapies.

**Abstract:**

Immunotherapies based on immune checkpoint blockade, neoantigen-reactive tumor-infiltrating lymphocytes and T cell receptor-engineered T cells (TCR-T) have achieved favorable clinical outcomes in tumor treatment. However, sustained immune response and tumor regression have been observed only in a few patients due to immune escape. Natural killer (NK) cells can mediate direct tumor lysis and target cancer cells with low or no expression of human leukocyte antigen class I (HLA-I) that are no longer recognized by T cells during immune escape. Therefore, the combination of T cell-based immunotherapy and NK cell therapy is a promising strategy for improving antitumor response and response rate. However, allogeneic NK cells for adoptive cell therapy have been limited by both the required cell number and quality. Here, we developed an efficient manufacturing system that relies on genetically modified K562 cells for the expansion of high-quality NK cells derived from peripheral blood mononuclear cells. NK cells with the optimal expansion and activity were identified by comparing the different culture systems. Furthermore, we demonstrated that the cooperation of NK cells with tumor-reactive T cells or with NY-ESO-1-specific TCR-T cells further enhanced tumors lysis, especially against tumors with downregulated HLA-I expression. The advantages of HLA-mismatch and non-rejection by other allogeneic immune cells demonstrated the potential of “off-the-shelf” NK cells with the capacity to target tumors for immunotherapy. Our results indicate that the combination strategy based on T cell and allogeneic NK cell immunotherapy might have potential for overcoming the barrier of immune incompetence caused by HLA-I downregulation.

## 1. Introduction

Natural killer (NK) cells are the cytotoxic lymphocytes of the innate immune system and are responsible for immune surveillance. The cytotoxicity of NK cells is controlled by a repertoire of active and inhibitory receptors expressed on their surface. Activating receptors, such as NKG2D and CD16, and natural cytotoxicity receptors, including NKp30, NKp44 and NKp46, transmit activation signals by inducing the phosphorylation of immunoreceptor tyrosine-based activation motifs contained within adapter proteins. On the other hand, inhibitory receptors, such as the killer immunoglobulin-like receptors (KIRs) and NKG2A, provide immunological self-tolerance and counteract the activity of NK cells when binding to their ligands, which are generally human leukocyte antigens class I (HLA-I) [1]. As the critical lymphocytes for cancer immunosurveillance, NK cells exert their cytotoxicity through similar mechanisms with CD8^+^ cytotoxic T cells but in a non-major histocompatibility complex (MHC)-restricted manner [2]. The isolation of NK cells from peripheral blood mononuclear cells (PBMCs) is one of the most easily available and common sources of primary NK cells. However, the primary NK cells are usually in a resting state and account for only a small proportion, averaging 10–15% of lymphocytes in peripheral blood, far below the quantities demanded for clinical adoptive NK cell therapy. The stimulation of cytokines, such as IL-2, IL-15, IL-21, IL-18 and IL-12, is required but insufficient for clinical-grade NK cell expansion, resulting in modest NK expansion capacity when used alone [3]. Using artificial antigen presenting feeder cells, such as autologous PBMCs, Epstein–Barr virus-transformed lymphoblastoid cells and K562 cells, has been shown to be more effective in inducing the large-scale expansion of NK cells [4,5,6,7]. In this study, we constructed a system using genetically modified K562 feeder cells stably expressing membrane-bound IL-21 (mIL-21), CD64, CD86 and 4-1BBL to obtain a sufficient number of activated NK cells and explored the effect of adding feeder cells or different media to the proliferation and activity of NK cells.

Immune checkpoint blockade therapies, such as anti-PD1/PDL1 and anti-CTLA4, which block the inhibitory signals of T cells activation, have demonstrated promising advances in different types of malignancies, but are still limited to a fraction of patients [8,9,10]. Furthermore, the adoptive transfer of the neoantigen-reactive tumor-infiltrating lymphocytes (TILs) and T cell receptor-engineered T cells (TCR-T) to redirect the antigen specificity against tumors have been effective in the treatment of certain cancer types [11,12]. NY-ESO-1 has been considered to be one of the most promising targets for TCR-T cells and is expressed by multiple tumors, including melanoma and carcinomas of lung, liver, gastric, prostrate and ovary [13]. However, these T cell-based immunotherapies are MHC-restricted and rely on HLA to present and recognize antigens. The clinical efficacy of neoantigen-reactive TILs or TCR-T therapy is primarily attributed to the effective CD8^+^ T cell responses that appear to target predominantly tumor-specific antigens exposed by HLA-I on the cell surface [14,15]. Tumors can exploit multiple escape mechanisms to evade immune recognition, including the regulation of antigen expression and HLA-I surface levels, alterations in the antigen processing and presentation machinery in tumor cells [16,17]. Evidence has indicated that tumors developed mutations in β2-macroglobulin (B2M) or transporters associated with antigen processing (TAP) proteins to limit the HLA-I presentation of antigens and facilitate their evasion from CD8^+^ T cells-mediated cytotoxicity [18,19,20]. Complete loss of HLA-I is a common event in several human tumors including melanoma, diffuse large cell B cell lymphoma (DLBCL), colorectal, lung, prostate, and renal cell carcinoma [21,22,23]. B2M deficiency that leads to the loss of HLA-I and eliminate antigen recognition by antitumor CD8^+^ T cells is likely a common mechanism of resistance to therapies targeting CTLA4 or PD1. Studies have shown that resistance through B2M loss can be overcome by an IL-2 pathway agonist potently activating an antitumor NK cell response as it is not restricted by HLA-I [24,25,26,27]. Thus, strategies aimed at targeting cells with low or no HLA-I should preferably be combined with T-cell therapy to maximize the likelihood of therapeutic benefits.

The adoptive transfer of NK cells is a promising adjuvant strategy and has shown its remarkable efficacy in cancer immunotherapy. The current innovative strategies of NK cell-based immunotherapy mainly focus on optimizing the proliferation and persistence of activated NK cells and improving the efficacy of therapeutic NK cells in vivo [28,29]. In recent years, some NK cell-based therapies, such as chimeric antigen receptor (CAR)-engineered NK cells and immune cell engagers between NK cells and tumor antigens have been translated into clinical-grade platforms and have yielded encouraging results in hematological tumors. By contrast, the current landscape of using NK cells as therapeutic agents for solid tumors is much less promising [30,31,32]. Various therapeutic combinations, genetic engineering approaches, alternative resources of NK cells, and other technologies are aiming at developing prospective NK cell antitumor therapies [32,33,34]. HLA-I downregulation or loss (“missing-self”) phenotypes would promote NK cell-mediated cytotoxicity. NK cells have therapeutic potential in the setting in which T cells cannot recognize cancer cells as a result of HLA-I downregulation. Therefore, we can take advantage of their immune surveillance function via the “missing-self” mechanism to combine allogeneic NK cells with T cells in cancer immunotherapy. Our data showed that the NK cells expanded by using our system were highly functional with broad cytotoxic activity against tumors and that allogeneic NK cells can be provided as an “off-the-shelf” cellular therapy to augment the antitumor reactivity of adoptively transferred T cells.

## 2. Materials and Methods

### 2.1. Cells and Culture

K562 (ATCC^®^CCL-243™), NALM6 (ATCC^®^CRL-3273™), THP-1 (ATCC^®^TIB-202™), HuT78 (ATCC^®^TIB-161™), A375 (ATCC^®^CRL-1619™), 786-O (ATCC^®^CRL-1932™), A549 (ATCC^®^CCL-185™) and HEK-293T cell lines (ATCC^®^ ACS4500™) were obtained from the American Type Culture Collection (ATCC, Manassas, VA, USA). Primary human PBMCs were obtained from MT-BIO (Shanghai, China), and the information on HLA allele is provided in Appendix A. The K562, NALM6, THP-1 and 786-O cells were cultured in RPMI1640 medium (11875093, Thermo Fisher Scientific, Waltham, MA, USA). The medium for 786-O cells required the addition of 1% L-glutamine (25030081, Thermo Fisher Scientific, Waltham, MA, USA) and 1% sodium pyruvate (11360070, Thermo Fisher Scientific, Waltham, MA, USA). A375, A549, and HEK-293T cells were cultured in DMEM medium (11995065, Thermo Fisher Scientific, Waltham, MA, USA). HuT78 cells were cultured in IMDM medium (12440053, Thermo Fisher Scientific, Waltham, MA, USA). All culture media were supplemented with 10% fetal bovine serum (FBS) (10099-141, Thermo Fisher Scientific, Waltham, MA, USA), 100 units/mL penicillin and 100 µg/mL streptomycin (15140122, Thermo Fisher Scientific, Waltham, MA, USA). All cells used for killing assays were transduced with luciferase-GFP lentivirus, and the purity of 95–100% was used for subsequent experiments.

### 2.2. Cytokines and Antibodies

Recombinant human IL-2 (78036.3, StemCell Technologies, Vancouver, BC, Canada) and IL-15 (200-15, PeproTech, Rocky Hill, NJ, USA) were used to expand NK cells. The modified K562 cells were stained with anti-4-1BBL-PE, anti-CD64-PECY7, anti-CD86-PE and anti-IL-21-APC. The purity and phenotypes of expanded NK cells were identified by staining with anti-CD56-PE, anti-CD3-APC, anti-NKG2D-PECY7, anti-CD16-FITC, anti-NKG2A-APC, anti-NKP30-BV421, anti-LAG-3-PE, anti-TIM-3-PECY7 and anti-PD-1-APC. For the knocking out of tumor cells in the TAP or B2M gene, we used anti-HLA-A, B, C-PE to detect the expression of HLA-I. For functional assays with NK cells or T cells, anti-CD8-APC/PE, anti-CD137-APC/PE, anti-IFN-γ-PE and anti-CD107a-APC/PE were used. Isotype-matched antibodies served as negative controls. All of these antibodies were purchased from Biolegend (San Diego, CA, USA). Cells were harvested and washed with FACS buffer (PBS containing 2% FBS), and then stained with the corresponding antibodies in the dark for at least 30 min at 4 °C before detection. After washing with FACS buffer, flow cytometric data were collected using an LSR II flow cytometer (BD Biosciences, San Jose, CA, USA) and analyzed using FlowJo software V10. For the measurement of IFN-γ production, cells were fixed and permeabilized by Cyto-Fast Fix/Perm Buffer Set (Biolegend, San Diego, CA, USA) and intracellular staining was performed according to the manufacturer’s instructions.

### 2.3. Plasmid Construction and Lentivirus Production

Human cDNA molecules CD64 (FcγRI, GenBank accession no. BC032634), CD86 (B7-2, GenBank accession no. NM_006889), and CD137L (4-1BBL, GenBank accession no. NM_003811) were cloned into a lentiviral expression vector that also encoded puromycin (pLVX-IRES-PURO). To generate mIL-21, the GM-CSF signal peptide (GenBank accession no. E01817.1) was directly liganded to the coding sequence of human IL-21 (GenBank accession no. NM_021803.4), and then the human IgG4 hinge region (GenBank accession no. HI539511.1), human IgG4Fc fragment (UniProtKB no. P01861) and human CD4 transmembrane domain (UniProtKB no. P01730) were sequentially fused in the frame. These sequences were codon-optimized on the website (https://sg.idtdna.com/CodonOpt, 25 October 2020). The mIL-21 was cloned into a lentiviral expression vector that encodes neomycin (pLVX-IRES-NEO). The optimized sequence of NY-ESO-1157-165 (SLLMWITQC) specific, HLA-A2-restricted 1G4 TCR was also inserted into the lentiviral vector pLVX-IRES-PURO. The lentivirus was produced by co-transfection into 293T cells with the psPAX2 packaging plasmid and the pMD2.G envelope plasmid.

### 2.4. Real-Time Quantitative Polymerase Chain Reaction (qRT-PCR)

Total mRNA was extracted by EZ-press RNA purification kit (No. B0004D, EZbio science, Shanghai, China), and first-strand cDNA was synthesized using the Maxima H Minus Reverse Transcriptase (EP0752, Thermo Fisher Scientific, Waltham, MA, USA). The qRT-PCR system was prepared using SYBR Green I Master Mix (04913914001, Roche, Basel, Switzerland). The relative expression level of each gene was normalized to GAPDH, and relative mRNA expression (fold change) for each gene tested was calculated as 2-ΔΔCT. The sequences of primers are listed in Appendix A.

### 2.5. Ex Vivo Expansion of NK Cells from Human PBMCs

To assist with NK cell expansion, K562 cells were genetically modified to express CD64, CD86, 4-1BBL and mIL-21. The K562 cells were seeded into 24-well plates at 2 × 10^5^ cells/well and incubated in 500 μL growth medium before infection. The lentiviral particles were added to the wells with 8 μg/mL polybrene for more than 72 h. After a 24 h incubation, the virus-containing medium was removed and replaced with 1 mL of fresh culture medium. After detecting the expression of the proteins, puromycin and neomycin were used to maintain the transfected cells. To determine the optimal culture conditions, modified K562 cells were irradiated (100 Gy) and then co-cultured with PBMCs at a 1:5 ratio in the RPMI1640 or NK MACS^®^ Medium (130-114-429, Miltenyi Biotec GmbH, Bergisch Gladbach, Germany). All media were supplemented with 10% human AB serum (100-512, Gemini, Woodland, CA, USA) and 10 IU/mL IL-2. Then, a half-volume of the medium was changed with fresh medium containing 20 IU/mL IL-2. At day 7, the concentration of IL-2 was raised to 300 IU/mL and 20 ng/mL IL-15 was added to the medium with irradiated modified K562 cells for restimulation. Total cell expansion and the purity of NK cells (stained with anti-CD3 and anti-CD56) were detected every 2 days after the 7th day by flow cytometry. Expanded NK cells were collected and cytotoxicity was tested against different tumor cells on day 14.

### 2.6. Cytotoxicity Assay and CD107a Degranulation

For the cytotoxicity test, 5 × 10^3^ target tumor cells (50 μL) were placed in a 96-well white flat bottom plate with triplicates and then co-cultured with corresponding ratios of effector cells (50 μL). The plates were centrifuged at 500 rpm for 3 min and then incubated for 24 h or 48 h in the CO_2_ incubator. Samples were treated with D-luciferin (88292, Thermo Fisher Scientific, Waltham, MA, USA) for 5 min prior to measurement.

To assess the degranulation of expanded NK cells or T cells against target cells, 5 × 10^5^ NK cells or T cells were co-cultured with target cells at a 2:1 ratio for 6 h in a 96-well U-bottom plate and anti-CD107a antibody was added at the first. Golgi-plug (1:1000, 555029, BD Biosciences, San Jose, CA, USA) and Golgi-stop (1:1500, 554724, BD Biosciences, San Jose, CA, USA) were added after 1 h and then co-cultured for the additional 5 h. Cells were washed and stained with other antibodies for flow cytometry analysis.

### 2.7. Enzyme Linked Immunosorbent Assay (ELISA)

The concentrations of IFN-γ (88-7316-88, Invitrogen, Carlsbad, CA, USA) in supernatants were measured by ELISA according to the manufacturer’s instructions.

### 2.8. Gene Editing by CRISPR/Cas9

To knock out the TAP1/TAP2 or B2M gene of the tumor cell lines, we designed the small guide RNA (sgRNA) from the Zhang lab at the Massachusetts Institute of Technology (http://crispr.mit.edu/, 12 August 2021). The sequences of sgRNA are listed in the Appendix A. The Cas9 protein (Z03469, YSY Biotech, Nanjing, China) was stored in Cas9 RNP buffer at −80 °C. Cas9 RNP was transduced into cells by electroporation. Briefly, Cas9 protein (5 ug) and sgRNA (2 ug) were mixed and then incubated for 10 min at room temperature. Subsequently, cells resuspended with Opti-MEM (31985070, Thermo Fisher Scientific, Waltham, MA, USA) were mixed with the assembled Cas9 RNP and prepared for electroporation. The editing efficiency of the B2M gene was detected by flow cytometry analysis using anti-B2M-PE (316306, Biolegend, San Diego, CA, USA), and the TAP1/2 gene was determined by the tracking of indels by decomposition (TIDE) assay. The primer sequences used to amplify TAP1 and TAP2 genomic regions are displayed in Appendix A. PCR samples were sequenced using the Sanger method and analyzed using the TIDE web tool (https://tide.nki.nl/, 15 November 2021).

### 2.9. Generation of 1G4 TCR-T Cells and Tumor-Reactive T Cells

To generate 1G4 TCR-T cells specific for NY-ESO-1 antigen, PBMCs were activated by CD3/CD28 magnetic beads (11131D, Thermo Fisher Scientific, Waltham, MA, USA) and cultured by ImmunoCult-XF T Cell Exp Medium (10981, StemCell Technologies, Vancouver, BC, Canada) with IL-2 (300 IU/mL). After 48 h, 1 × 10^6^/mL T cells were plated in 24-well plates and transduced with the 1G4-TCR lentivirus. Then, cells were expanded for 5 days and TCR Vβ13.1-PE (IM2292, Beckman Coulter, Miami, FL, USA) was used to assess the transduction efficiency of 1G4-TCR. On day 12, 1G4 TCR-T cells were harvested.

Tumor-reactive T cells were obtained from the co-culture of PBMCs with tumor cells. The 24-well plates (3524, Corning Incorporated, Corning, NY, USA) were coated with 5 μg/mL anti-CD28 (14-0289-82, eBioscience, San Diego, CA, USA) and kept overnight at 4 °C. Anti-CD28-coated plates were washed with PBS and PBMCs were seeded at 10^6^ cells/well and stimulated with A375, A549 and 786-O cells, respectively, at an effector:target (E:T) ratio of 15:1. The culture media was composed of T Cell Exp Medium, 10% human AB serum, 300 IU/mL IL-2 and 20 μg/mL anti-PD-1-blocking antibody (A2002, Selleck Chemicals, Houston, TX, USA). Half of the medium was refreshed every two days. On day 5, cells were collected and restimulated with fresh tumor cells at a 20:1 ratio. On day 10, cells were collected for evaluation of tumor reactivity and killing assays.

### 2.10. RNA Sequencing

Total RNA was extracted from NK cells expanded with different culture systems. The concentration and purity of isolated RNA were measured with NanoDrop-2000 spectrophotometer. RNA libraries were constructed with Illumina Truseq™ RNA sample prep Kit. Sequencing was carried out using a 2 × 150 bp PE configuration. The clean data (reads) were mapped using Hisat2 (version 2.1.0). Then, we performed gene expression analysis through RNA-seq by Expectation Maximization (RSEM, version 1.3.3). |log2FC| ≥ 1 and false discovery rate (FDR) < 0.05 were determined as thresholds to screen different expression genes (DEGs). Venn analysis was used to calculate the number of genes in each gene set and show the overlap relationship among different gene sets. Gene ontology (GO) analysis displayed the most significant biological process of a particular gene set (q < 0.05) according to the filtered DEGs. Biological pathway terms were enriched (q < 0.05) based on the Koto Encyclopedia of Genes and Genomes (KEGG) pathway database. To validate the reliability of the DEG results, the expression levels of selected transcripts were determined by qRT-PCR with GAPDH as an endogenous reference.

### 2.11. Statistical Analysis

Quantified data were expressed as mean ± SD. Student’s t-test was used for the comparison between the two groups. One-way analysis of variance (ANOVA) or two-way ANOVA with correction for multiple comparisons was used when appropriate. All statistical analyses were performed with GraphPad Prism 9.0.

## 3. Results

### 3.1. NK Cells Expanded Robustly from PBMCs with Genetically Engineered K562 Cells

K562 cells have already been reported to be a favorable choice as feeder cells for NK cell expansion. However, it is difficult to achieve large-scale expansion without additional co-stimulatory factors [35,36]. To obtain the large-scale highly cytotoxic NK cells needed for clinical adoptive therapy, we utilized K562 cells expressing co-stimulatory molecules and membrane-bound cytokine as artificial feeder cells. Genes CD64, CD86, and 4-1BBL, which encode ligands that had previously been identified as important factors capable of stimulating massive NK cell proliferation [37,38], were transduced. IL-21 has been described as a prospective cytokine that promotes the antitumor potential of NK cells [39]. Referring to the technology of Cooper et al. [40], we designed a novel structure of IL-21 bound to the cell surfaces of K562 cells, which is able to replace the need for soluble IL-21 in order to continuously stimulate NK cell proliferation. The CD64, CD86 and 4-1BBL coding sequences were constructed in the lentiviral vector PLVX-IRES-PURO, and the mIL-21 coding sequence was constructed in the lentiviral vector PLVX-IRES-NEO (Figure 1A). Lentivirus were produced and then transduced into K562 cells. Antibiotic selection was used to identify and maintain transfected cells. The messenger RNA (mRNA) levels of CD64, CD86, 4-1BBL and mIL-21 were significantly increased in the transduced cells compared with parental K562 cells (Figure 1B). Accordingly, flow cytometry analysis proved that modified K562 cells had high expressions of CD64, CD86, 4-1BBL and mIL-21, compared with the conventional K562 cells (Figure 1C).

To seek a better condition for the ex vivo expansion of primary NK cells than in previously reported protocols, we investigated the effects of different culture mediums and the addition of feeder cells on the activity of NK cells. The expansion procedure of NK cells from PBMCs is displayed in Figure 1D. To verify whether our modified K562 cells were able to promote NK cell expansion, we compared culture groups that use the RPMI1640 or NK MACS medium in the presence or absence of modified K562 cells. We found that when using RPMI1640 medium without the addition of modified K562 cells, NK cell expansion was limited and cells were progressively disintegrated. Identical concentrations of IL-2 and IL-15 were used for each culture group. The purity and fold expansion of NK cells in three culture groups were detected during the 14 days (Figure 1E,F). Co-culture with our modified K562 feeder cells dramatically improved the purity and quantity of NK cells. The purity of NK cells expanded in the group of NK MACS medium with modified K562 cells (MACK, red) and the group of RPMI1640 medium with K562 cells (RPMIK, blue) were almost close to 98% at the day 14. However, the expansion of NK cells in the MACK group was significantly higher than that in the groups of NK MACS medium without modified K562 cells (MAC, green) and RPMIK. We also detected the phenotypes of the expanded NK cells on day 14 through flow cytometry analysis (Appendix A).

### 3.2. NK Cells Expanded from NK MACS Medium with Modified K562 System Performed the Strongest Cytotoxic Activity

To compare the activity of NK cells expanded in the three groups, we investigated the direct cytotoxicity by co-culturing with different solid or hematological tumors at an E:T ratio of 2:1 for 24 h (Figure 2A), and detected CD107a surface expression and the IFN-γ production of NK cells after incubation for 5 h at an E:T of 5:1 (Figure 2B,C) (the K562 target cell was used as a positive control because of its apparent sensitivity to NK cells due to the lack of HLA-I expression). Expanded NK cells from the MACK group displayed the strongest cytotoxicity against all the experimental tumor cell lines, including NALM6, THP-1, HuT78, A375, 786-O and A549. Additionally, the HLA-I expression of the tested tumor cell lines is shown in the Appendix A. In conclusion, the simple use of NK MACS medium showed the limited expansion and low cytotoxicity of expanded NK cells, but co-culture with modified K562 cells was able to improve the NK cell activity. Although NK cells from the RPMIK group had favorable fold expansion and purity, their cytotoxicity was significantly lower than that from MACK group. Our data showed that PBMCs stimulated by NK MACS medium with modified K562 system were able to reach a nearly 4000-fold expansion and were able to obtain NK cells with high purity and optimal cytotoxicity after culturing for 14 days.

### 3.3. RNA Sequencing Revealed That Expanded NK Cells from Different Culture Systems Exhibited Diverse Transcriptional Landscapes

To further explain difference in activity among NK cells derived from different culture conditions, the comprehensive transcriptome profiling of the expanded NK cells from different culture groups was performed by RNA-sequencing (RNA-seq). We selected NK cell samples expanded from two different donors for RNA-seq analysis, focusing on determining the gene expression differences among NK cells from the same donor cultured by different methods mentioned above. The principal component analysis (PCA) of expanded NK cell samples exhibited distinct gene expression properties of these cell populations (Figure 3A). The closer the distance between each sample point, the higher the similarity between samples, which also indicates the biological repetition of each group. PC1 accounted for 27% of the transcriptional differences and suggested that NK cells expanded from different donors had a heterogeneity of the gene expression. PC2 accounted for approximately 14% of differences that were mainly exhibited amongst the NK cell samples under different culture conditions.

Compared with NK MACS medium alone, the expansion of NK cells with genetically modified K562 cells lead to an upregulation of 120 genes and a downregulation of 248 genes, using a threshold of |log2FC| ≥ 1 and adjusted *p* value < 0.05 assigned by the DESeq2 algorithm. Additionally, both were stimulated by modified K562 cells, using NK MACS medium lead to a significant upregulation of 815 genes and downregulation of 540 genes compared with using RPMI1640 medium (Figure 3B, Appendix A). By assessing the involved biological process of these DEGs through GO term enrichment analysis, we found that DEGs in the MACK group were mainly enriched in terms of the regulation of cell killing, the positive regulation of cell activation and production of molecular mediator of immune response when compared with MAC group. The DEGs in NK cells from the MACK group were mainly enriched in the positive regulation of lymphocyte proliferation and the positive regulation of production of molecular mediator of immune response, compared with those in the RPMIK group (Figure 3C, Appendix A).

On the basis of direct cytotoxicity against tumor cells, CD107a surface expression and IFN-γ production, we concluded that NK cells expanded by the MACK culture system showed better activity than those expanded by the other methods. Therefore, we sought the 128 common-DEGs (CO-DEGs) among NK cells from the MACK group in comparison with the other two groups, respectively, and displayed the results via Venn diagram (Figure 3D, Appendix A). To analyze the CO-DEGs’ contribution to the transcriptome characteristics of NK cells equipped with strong cytotoxicity, we performed the GO enrichment and KEGG pathway enrichment analysis on the CO-DEGs (Figure 3E). The CO-DEGs were principally enriched in the GO term of the regulation of leukocyte proliferation, the regulation of cell killing, cytokine receptor binding, the positive regulation of cytokine production and the regulation of cell–cell adhesion, which were closely related to the expansion, persistence and cytotoxicity of NK cells. Moreover, KEGG pathway analysis demonstrated that these CO-DEGs primarily participated in certain pathways related to the activity of NK cells, such as cytokine–cytokine receptor interaction, the NF-kappa B signaling pathway, NK cell mediated cytotoxicity, the TNF signaling pathway and the JAK-STAT signaling pathway. The heatmap (Figure 3F) shows the CO-DEGs associated with the pathway of NK cell mediated cytotoxicity in the three different culture groups. We observed that expanded NK cells with strong cytotoxicity highly expressed genes such as IFNGR1, NKP44, NKP46, NKG2D, NKG2E, 2B4, CD3ζ and FCGR but lowly expressed KIRS. The expression levels of nine candidate CO-DEGs (upregulated: NKG2D, CD16a, NKP44, NKP46 and IFNGR1; downregulated: KIR3DL1, KIR3DL2, KIR3DL3, and KIR2DL1) were verified by qRT-PCR (Figure 3G), in accordance with the RNA-seq results.

### 3.4. Tumor Cells with Low Expression Level of HLA-I Were More Sensitive to NK Cells

Tumor-reactive T cells are key mediators of tumor destruction. However, the loss or downregulation of HLA-I molecules is commonly observed during tumor progression, resulting in the impairment of HLA-restricted cytotoxic T lymphocytes-mediated tumor immunity [41]. As B2M is a component of all HLA-I allotypes, a heterozygous deleterious mutation at B2M can lead to the reduced surface expression levels of HLA-I [18]. TAP-deficient cells cannot transport peptides to the endoplasmic reticulum, reducing the availability and variety of peptides accessible for binding to HLA-I. The reduction in peptide binding also reduces the stability and surface expression of HLA-I on TAP-deficient cells [20]. Focusing on the possible functional changes of the HLA-I complex, including deep deletions, missense mutations and truncating mutations that are expected to truncate the protein, we performed an analysis of The Cancer Genome Atlas (TCGA) database based on cBioportal (www.cbioportal.org/, 16 October 2022). An oncoprint plot showed genomic alterations in genes encoding the HLA-I, B2M, TAP1 and TAP2 that would influence antigen presentation, amounting to ~12% of 10,967 cancer patients in the TCGA data set (Figure 4A). TAP1 and TAP2 showed concerted behavior in genomic alterations with HLA-I. Green bar graphs at the top showed total somatic mutation counts and a microsatellite instability score (MSIsensor score). In addition, the distribution of these genomic alterations in different TCGA studies was displayed according to the PanCancer data analysis. DLBCL showed the highest frequency of mutations and depletion (Figure 4B). Understanding the prevalence of these events is important for estimating the downregulation of surface HLA-I expression and the loss of antigen presentation and for designing appropriate immunotherapy strategies for different cancers that escape immune recognition. Therefore, we investigated whether the immune surveillance function of NK cells, which have potent activity against HLA-I-downregulated tumor cells, can be used to compensate for tumor resistance to specific T cells.

To simulate the immune evasion mechanism of HLA-I downregulation in tumor cells in vitro, we first knocked out TAP1 and TAP2 simultaneously and knocked out B2M genes in several solid tumor cells (A375, A549 and 786-O) using CRISPR/Cas9 technology, respectively. Then, the efficiency of TAP gene editing was assessed by TIDE analysis, and the efficiency of B2M gene editing and the expression of HLA-I were determined by flow cytometry (Figure 4C,D and Appendix A). Luciferase-based cytotoxicity assays showed that tumor cells with knocked out TAP or B2M were obviously lysed compared with primary tumor cells (control), while incubation with NK cells at each different ratio (Figure 4E,F), indicated that tumor cells with downregulated HLA-I expression were more susceptible to the cytotoxicity of NK cells.

### 3.5. Generation and Identification of Tumor-Reactive T Cells

Tumor antigen stimulation leads to the clonal expansion of tumor-reactive T cells. We referred to and validated the method of Dijkstra et al. to obtain the tumor-specific T cells from PBMCs co-cultured with tumors [42] and then analyzed the response of tumor-reactive T cells with matched tumor cells. We utilized three solid tumor cells during the attempt to obtain tumor-reactive T cells. Briefly, plate-bound anti-CD28 was used to provide co-stimulation and IL-2 was added to support T cells proliferation. In addition, we added blocking antibody to PD-1 to counteract the inhibitory effect of the PD1/PDL1 pathway and alleviate exhaustion during T cell activation. PBMCs were stimulated every 5 days with fresh tumor cells (Figure 5A). Tumor response by CD8^+^ T cells was assessed after 10 days of co-culture, by detecting the production of IFN-γ, the degranulation marker CD107a and activation marker CD137 (Figure 5B–D). Stimulation with three different tumor cells, respectively, all induced CD137 and CD107a upregulation, and IFN-γ secretion in CD8^+^ T cells after 10 days of co-culture. We also evaluated the tumor reactivity of T cells before co-culture with tumor cells, and responses were not observed. The markedly improved tumor response after each round of stimulation indicated that our co-culture system can be used to successfully expand tumor-reactive T cell populations against tumors (Figure 5E).

### 3.6. Combination of NK Cells with Tumor-Reactive T Cells or NY-ESO-1-Specific 1G4-T Cells Effectively Improves the Cytotoxicity against Tumors

After confirming the feasibility of inducing tumor-reactive CD8^+^ T cells, we assessed the efficiency of tumor destruction by these T cells and determined their alloreactivity against wild-type or HLA-I-low tumors. Therefore, we co-cultured tumor-activated alloreactive T cells with matched A375, 786-O or A549 cells, and corresponding tumor cells with downregulated HLA-I after TAP or B2M gene knockout. T cells from the same donor activated by CD3/CD28 magnetic beads were used as a negative control (Figure 6A,B). The cytotoxicity of tumor-reactive T cells against tumor cells with downregulated HLA-I was weakened significantly. At the same time, the expanded NK cells were added to test whether the efficiency of tumor killing could be improved. The results demonstrated that the combination of tumor-reactive T cells with NK cells was able to directly enhance the killing of tumors, regardless of whether tumor cells downregulated the expression of HLA-I. NK cells that expanded from different conditions also showed different cytotoxicity against tumor cells with HLA-I downregulation when combined with T cells. NK MACS with modified K562 cells displayed optimal outcomes, which was consistent with our previous conclusions.

We obtained HLA-A2-restricted, NY-ESO-1-specific 1G4-T cells by transducing activated T cells with the 1G4-TCR lentivirus (Figure 6C). Wild type A375 cells demonstrated NY-ESO-1 and HLA-A2 expression as measured via flow cytometry (Appendix A). We investigated the cytotoxicity of 1G4 TCR-T cells against A375 and TAP-knockout A375 cells. Activated T cells without the transduction of 1G4-TCR served as control. Similarly, we used a combination of NK cells and 1G4 TCR-T cells to co-culture with tumor cells and compared their antitumor effects. The 1G4 TCR-T cells showed significantly impaired killing activity against A375 cells with downregulated HLA-I expression. After combination with NK cells, the killing efficiency of tumor cells was apparently increased (Figure 6D). Therefore, our results suggested that the combination of NK cells with tumor-activated alloreactive T cells or 1G4 TCR-T cells can effectively improve the cytotoxicity against tumors, especially for enhancing the control of tumors that are resistant to T cells due to the downregulation of HLA-I.

### 3.7. Allogeneic PBMCs Do Not Affect the Persistence and Cytotoxicity of NK Cells In Vitro

Earlier studies have proven that adoptively transferred human allogeneic NK cells derived from haploidentical related donors can be expanded in vivo [43,44]. The initial large-scale study of CAR-modified NK therapy demonstrated that no cases of graft-versus-host-disease were observed, despite the HLA mismatch between the patients and their CAR-NK products. The injected CAR NK cells were reported to persist in the blood for more than a year [45]. These studies showed that the administration of allogeneic NK cells is efficacious and safe, which encourages the potential of allogeneic NK cells to be used as an “off-the-shelf” product for the treatment of malignancies. To validate this, we further investigated whether the persistence and activity of our expanded NK cells could be affected by the allogeneic immune cells. Expanded NK cells from donor1 (Appendix A) were labeled with the dye eFluor™ 670 (65-0840, Invitrogen, Carlsbad, CA, USA) and then co-cultured with PBMCs from four different donors (donor 1 to 4) at a ratio of 1:1 for 5 days, respectively. The persistence of NK cells was detected every 24 h using a flow cytometer with a 660/20 bandpass filter (equivalent to APC). Figure 7A showed no significant change in the percentage of labeled NK cells within 5 days of co-culture with PBMCs. The histogram in Figure 7B depicts the labeled NK cells and PBMCs at day 5, and the mean fluorescence value of each group was calculated. The group of NK cells alone served as the control. During the 5 days of co-culture, we observed no rejection between allogeneic PBMCs and NK cells, and the persistence of NK cells was not affected.

To determine whether the cytotoxicity of universal allogeneic NK cells was affected when used alone or in combination with tumor-reactive T cells for different recipients, we performed experiments in vitro in which primary A375 or TAP-knockout A375 cells were cultured alone or in different combinations with NK cells (from donor N1 or N2), tumor-reactive T cells and PBMCs (both from donors P1-P4) for 48 h at an E:T ratio of 2:1. Additionally, the killing efficiency of each group was calculated and displayed (Figure 7C,D). As expected, the combination of NK cells and tumor-reactive T cells resulted in more effective cytotoxicity against tumor cells than NK cells or tumor-reactive T cells alone, especially against TAP-knockout A375 cells with downregulated HLA-I expression. With or without the combination of tumor-reactive T cells, allogeneic PBMCs did not reduce the effective cytotoxicity of NK cells against A375 and TAP-knockout A375 cells. We observed similar results with PBMCs and T cells isolated from four different donors, suggesting that NK cells were not rejected by allogeneic immune cells. The persistence and cytotoxicity of NK cells were not affected, further demonstrating the universal property of NK cells in combination with T cells for the treatment of different patients.

## 4. Discussion

Allogeneic NK cells are potential candidates for “off-the-shelf” cancer immunotherapy, because the risk of graft-versus-host disease and side effects, such as cytokine release syndrome and neurotoxicity, are absent. Unlike T cells, adoptive NK cells exert cytotoxicity against tumors without the requirement of HLA matching [46]. For further advancements in NK cell-based cancer immunotherapy, optimized culture systems are essential to reliably obtaining a large number of highly functional cells that can be cryopreserved and thawed for “off-the-shelf” use. In this study, we developed an alternative approach using our genetically modified K562 cells for the rapid expansion of primary NK cells from peripheral blood. The modified K562 cells expressing mIL-21, CD64, CD86 and 4-1BBL provide co-stimulatory signals that are vital to augmenting NK cell expansion to the clinical scale via cell-to-cell contact in the co-culture system. We compared the MACK expansion system with RPMIK and MAC system. The total cell number, purity and functionality of expanded NK cells from the MACK system were superior to those of NK cells from the other two approaches. NK cells from the MACK system showed a nearly 4000-fold expansion and exhibited a high cytotoxic function against diverse hematologic and solid tumor cell lines. The cell lines tested included leukemia cells, lung carcinoma cells, melanoma cells and renal carcinoma cells. RNA-seq data showed that the DEGs of NK cells obtained by different culture conditions were mainly associated with pathways related to NK activity, such as cytokine–cytokine receptor interaction, natural killer cell mediated cytotoxicity, the NF-kappa B signaling pathway, the TNF signaling pathway and JAK-STAT signaling pathway. NK cells from the MACK system exhibited strongly cytotoxic phenotypes, which also explained potential underlying mechanisms. The RNA-seq data provided in this study prompted the altered gene expression of NK cells expanded under different culture conditions and could pave the way for other studies to optimize NK cells with strong cytotoxicity.

A growing number of clinical studies have shown favorable safety profile and efficacy of NK cell therapies [30,47,48,49]. Serious toxicities, such as cytokine release syndrome and neurotoxicity, and graft-versus-host disease are absent after the infusion of allogeneic NK cells. Compared with T cells, the shorter lifetime and lesser production of cytokines by NK cells are the two major reasons for the decrease in the risk of side effects [50]. The NK cell killing of a target cell depends on the balance between inhibitory and activating signals. The tested tumor cell lines with HLA-I expression in our study are more sensitive to the expanded NK cells from the MACK system because of the upregulation of activating receptors and the downregulation of inhibitory ones on these NK cells. Genetically engineered NK cells that overexpress activating receptors or eliminate the expression of inhibitory receptors have consistently demonstrated enhanced antitumor effects with good safety profiles [51,52,53]. An earlier study has suggested that in interactions between NK cells and normal untransformed cells, MHC-I molecules are in most cases expressed in excess compared to what is functionally needed to ensure self-tolerance [54]. Therefore, the normal, autologous cells may be able to maintain a safety margin over a range of MHC-I expression levels, in order to ensure robustness in NK cell tolerance. Although NK cells expanded from the MACK system probably showed an augmented response owing to enhanced cytotoxicity against HLA-I-expressed tumor cells, the possibility of their toxicity to human normal cells should be very small. Additionally, the strength of NK cells as a safe and reliable host cell type for universal cell therapy is probably due to its natural properties as a key player of innate immune response. Recently, the transfer of TILs targeting tumor neoantigens and the engineering of blood T cells with tumor-specific TCRs have shown promising clinical outcomes in cancer immunotherapy [55,56,57,58]. However, TILs and TCR-T cells can only identify specific antigens presented by HLA molecules and may induce immune incompetence because of the downregulation or mutation of HLA in tumor cells; thus, their clinical applications are limited [59]. Previous studies on the mechanisms of response and resistance to T cell-based therapy have suggested that genetic defects that may influence responses to immunotherapy by inhibiting antigen processing and presentation [60]. Since the HLA-I molecule is not properly assembled in the absence of peptide and/or β2m, loss of HLA-I may depend on alteration of B2M expression or mutations of TAP1/2. HLA-I downregulation through the loss of heterozygosity or B2M mutations is an escape mechanism that may impede TCR gene therapy effectiveness [61,62]. TAP deficiency is generally associated with increased tumorigenicity, conceivably representing an immune evasion mechanism due to an overall reduction of antigen presentation to CD8^+^ T cells by the HLA-I molecules [20,63]. With the capacity of NK cells to specifically sense the absence of HLA-I, adoptive NK cell therapy may be an attractive rescue option for patients with failed conventional immunotherapy or with low level of HLA-I at diagnosis. In our experiments, HLA-I expression was significantly reduced in tumor cells by knocking out B2M or TAP genes. Combination therapies and strategies to promote bystander killing may be investigated for cancers with heterogeneous genetic defects. As we expected, the cytotoxicity of NK cells was remarkably enhanced against several tested tumor cells with downregulated HLA-I expression due to reduced TAP or B2M expression. In addition, we established and validated a system to induce and analyze tumor-specific T cell responses for several tumor cells in a personalized manner and demonstrated that the co-cultures of tumor cells and PBMCs can be used to enrich tumor-activated alloreactive T cells. HLAA2-restricted NY-ESO-1-specific 1G4 TCR-T cells were obtained by the transduction of 1G4-TCR lentivirus. In the cytotoxicity assays, both tumor-reactive T cells and NY-ESO-1-specific 1G4 TCR-T cells showed a decreased cytotoxicity against tumor cells with downregulated HLA expression.

Our data successfully supported the notion that NK cells that maintain strong antitumor function after cryopreservation have the ability to target cancer cells that have evaded T cells recognition through HLA downregulation. The persistence and cytotoxicity of NK cells were not affected by allogeneic immune cells, further demonstrating the “off-the-shelf” potential of NK cells. Therefore, combining “off-the-shelf” allogeneic NK cells with TCRs-dependent T cell therapy can effectively overcome the obstacles in the treatment of patients with tumors that have low HLA-I expression. Our research work on the association between NK cells and tumor-reactive T cells or TCR-T cells also represents a step towards future combination strategies for cancer immunotherapies. The limitation of our study is that the interaction between allogeneic NK and T cells were not illustrated. We presumed that tumor-infiltrating NK cells are important for both direct cytotoxicity and the modulation of the microenvironment to promote T cell recruitment and activation. Comprehensive research of this combination strategy needs to be further developed.

## 5. Conclusions

The large-scale expansion of NK cells with strong cytotoxic activity was obtained and transcriptional alterations occurring from expanded NK cells in different culture system were analyzed in this study. This work proved that combining “off-the-shelf” allogeneic NK cells with TCR-dependent T cell therapy can effectively improve tumor killing and overcome the obstacles in the treatment of tumors with low HLA-I expression. Such a combination immunotherapy provides a bright alternative for refractory tumors.

## Figures and Tables

**Figure 1 cancers-15-00251-f001:**
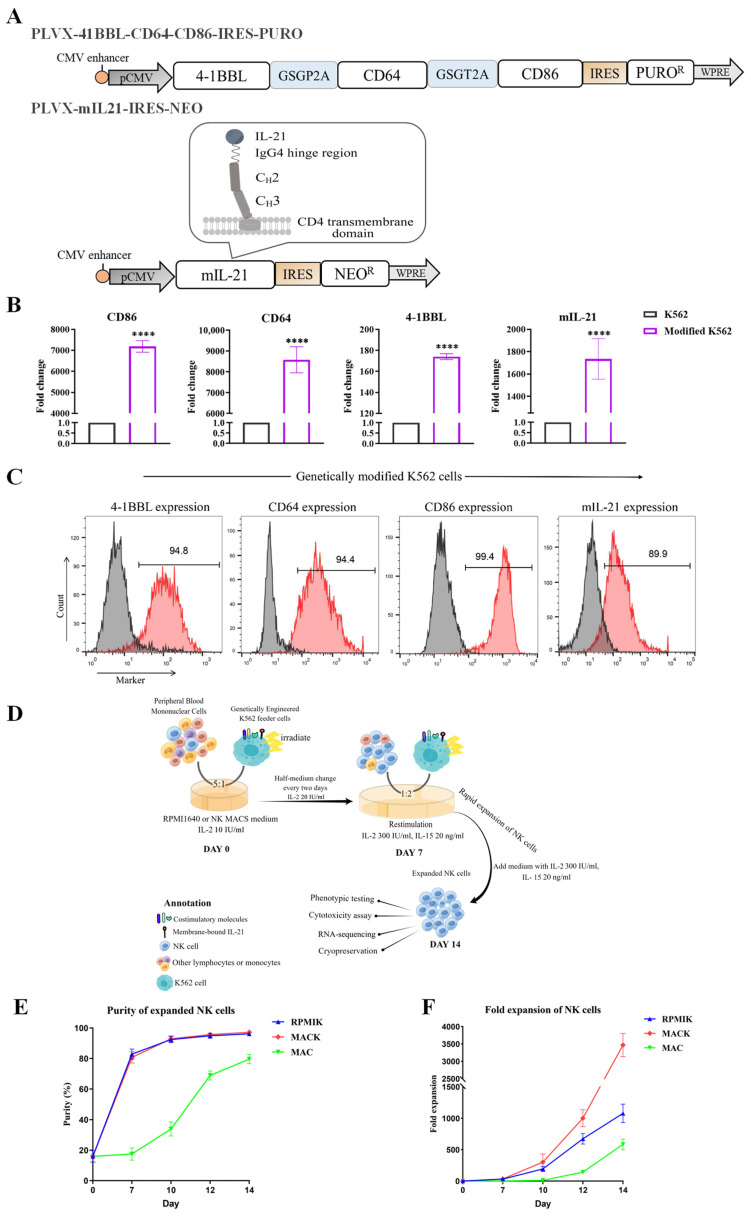
Expansion of primary NK cells from PBMCs with genetically engineered K562 feeder cells. (**A**) Schematic diagram of CD64, CD86 and 4-1BBL coding sequences in the lentiviral vector backbone PLVX-IRES-PURO and mIL-21 coding sequences in the lentiviral vector backbone PLVX-IRES-NEO, along with a box showing the folded mIL-21 protein. The coding sequences of CD64, CD86 and 4-1BBL are connected by P2A-GSG and T2A-GSG self-cleaving peptides that allow the simultaneous expression of three genes. (**B**) The mRNA levels of CD64, CD86, 4-1BBL and mIL-21 were determined by quantitative RT-PCR. GAPDH was taken as the house keeping gene and data were expressed as fold changes relative to control. *n* = 3. Statistical analysis by unpaired *t* test. **** *p* ≤ 0.0001. (**C**) The surface expression of CD64, CD86, 4-1BBL and mIL-21 on modified K562 cells was determined by flow cytometry analysis. (**D**) Procedure for NK cell manufacturing with K562 feeder cells. (**E**) The purity of expanded NK cells was detected by flow cytometry using anti-CD3-APC and anti-CD56-PE. (**F**) Fold expansions of NK cells in the three groups during 14 days of culture.

**Figure 2 cancers-15-00251-f002:**
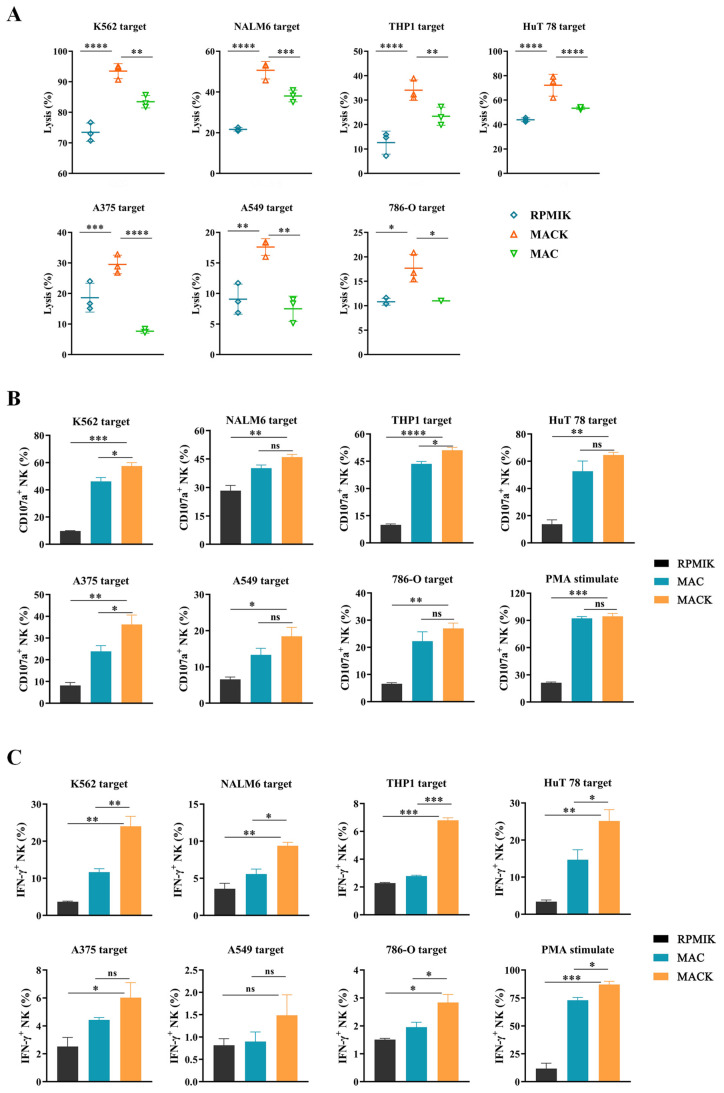
Comparison of the cytotoxic activity of NK cells obtained from different culture groups. (**A**) Luciferase-based cytotoxicity assays were performed in NK cells co-cultured with different solid or hematological tumors at E: T to 2:1 for 24 h (*n* = 3). (**B**,**C**) CD107a surface expression (**B**) and IFN-γ production (**C**) of NK cells after incubation with different tumor cells at a 5:1 ratio for 5 h are displayed in histograms. Statistical analysis by one-way ANOVA with Dunnett’s correction for multiple comparisons. * *p* ≤ 0.05; ** *p* ≤ 0.01; *** *p* ≤ 0.001; **** *p* ≤ 0.0001.

**Figure 3 cancers-15-00251-f003:**
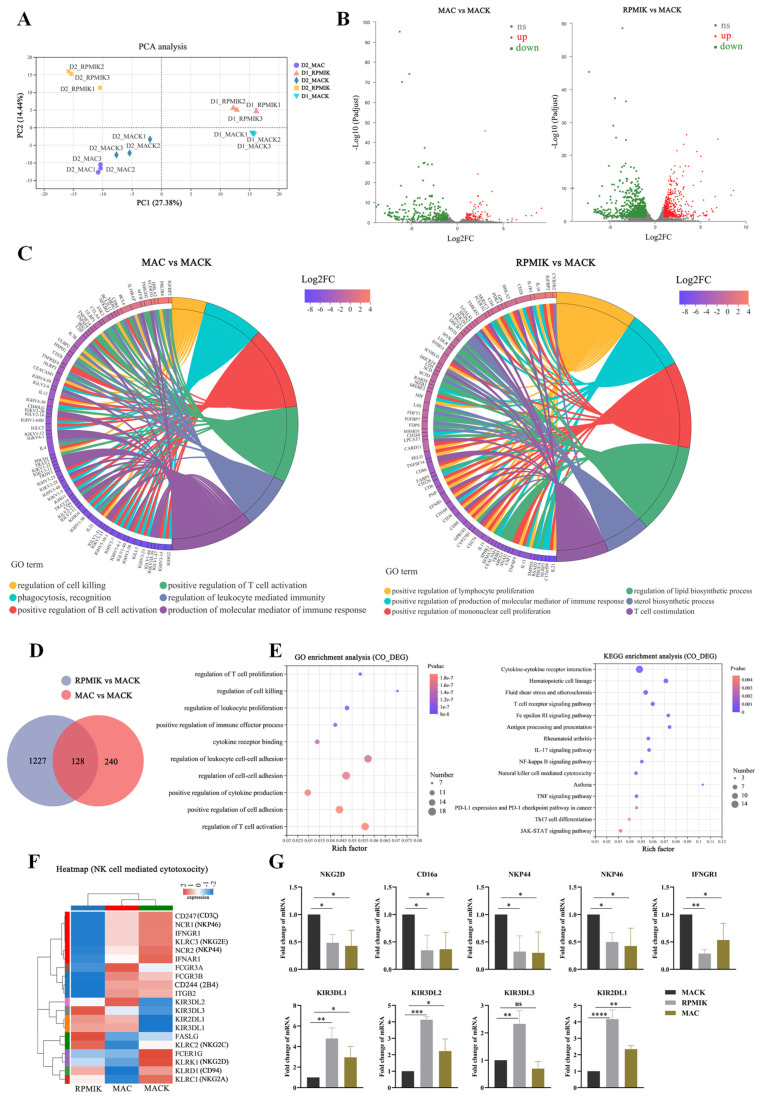
RNA expression profiling of the expanded NK cells from different culture groups. (**A**) PCA of the expanded NK cell samples from different culture conditions; each group has three biological replicates (D1 = Donor1, D2 = Donor2). (**B**) Volcano plots to illustrate the distribution of significance (−log10 (adjusted *p* value)) and change in expression (log2(fold change)) of DEGs within the comparisons. (**C**) GO term enrichment analysis of DEGs. The top 6 significantly enriched biological processes of these DEGs within the two comparisons are shown. (**D**) Venn diagram shows the number of common differentially expressed genes (co-DEGs) of the comparison groups. (**E**) The results of GO enrichment analysis (Top 10) and KEGG pathway enrichment analysis (Top 15) of CO-DEGs are shown in the bubble chart, respectively. (**F**) The pathways of NK cell mediated cytotoxicity associated CO-DEGs are displayed in the heatmap. (**G**) Validation of the DEGs by qRT-PCR. Statistical analysis by one-way ANOVA with Dunnett’s correction for multiple comparisons. * *p* ≤ 0.05; ** *p* ≤ 0.01; *** *p* ≤ 0.001; **** *p* ≤ 0.0001; ns represents not significant.

**Figure 4 cancers-15-00251-f004:**
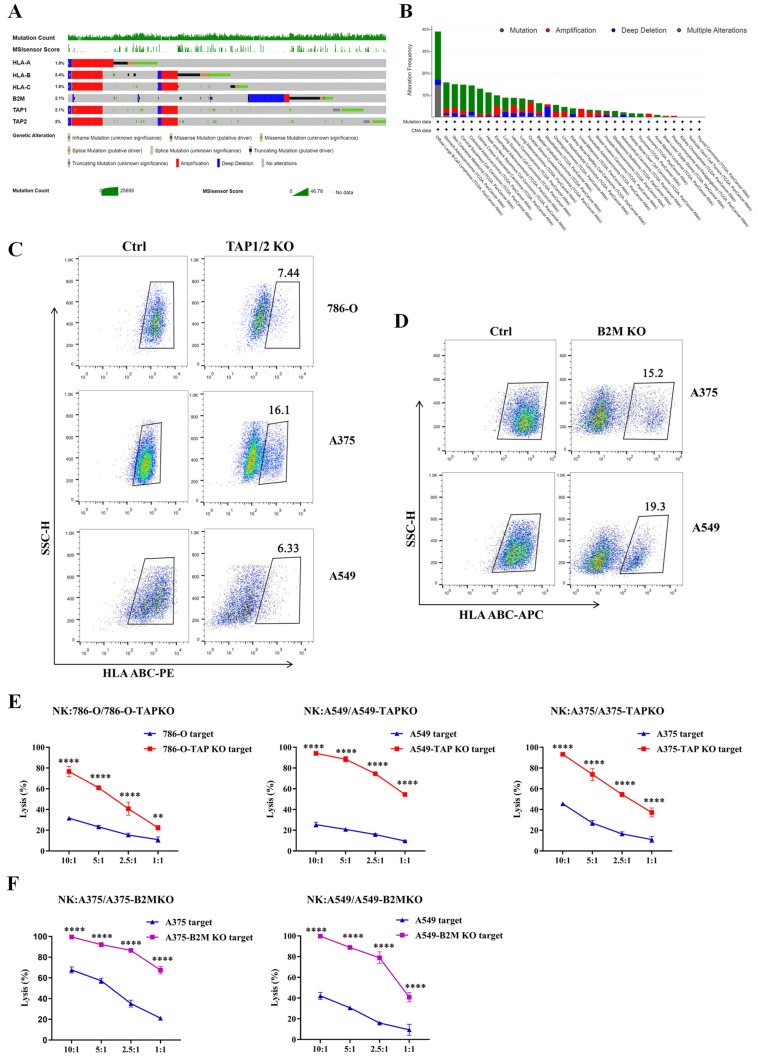
Immunosurveillance function of expanded NK cells against tumor cells with downregulated HLA-I expression. (**A**) An oncoprint plot using cBioportal shows genomic alterations in genes encoding the HLA-I, B2M, TAP1 and TAP2 across the range of TCGA datasets. (**B**) PanCancer data analysis provides an overview of the distribution of these genomic alterations in the different TCGA studies. DLBC has the highest frequency of mutations and losses. (**C**,**D**) Detection of the expression of HLA-I on tumor cells after knockout of TAP1/TAP2 (**C**) and B2M (**D**) molecules by flow cytometry analysis using anti-PE or APC-HLA A, B, C. (**E**,**F**) Enhanced cytotoxicity of the expanded NK cells against tumor cells with downregulated expression of HLA-I due to knockout of TAP (**E**) or B2M (**F**) gene. Statistical analysis by two-way ANOVA with Bonferroni’s correction for multiple comparisons. ** *p* ≤ 0.01, **** *p* ≤ 0.0001. All data were representative of at least three independent experiments.

**Figure 5 cancers-15-00251-f005:**
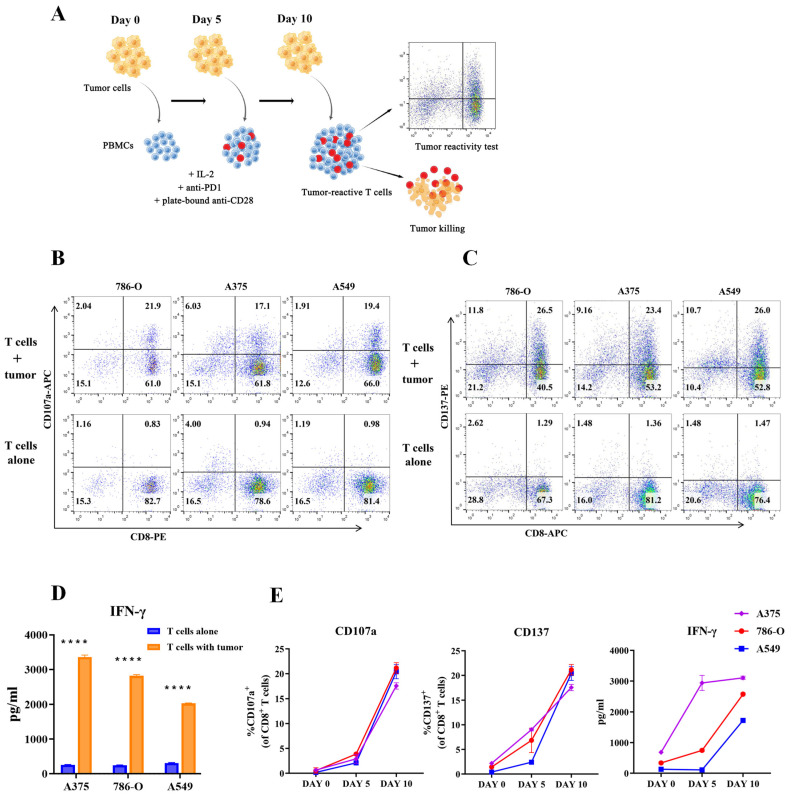
Generation and identification of tumor-reactive T cells. (**A**) The experimental workflow was displayed. PBMCs were stimulated every five days with fresh tumor cells. After 10 days of co-culture, CD8^+^ T cells reactivity and cytotoxicity were estimated. (**B**,**C**) Test for reactivity against tumors of CD8^+^ T cells after 10 days of co-culture with tumor cells and analysis of CD107a expression (**B**) and CD137 expression (**C**) gated on CD8^+^ T cells by flow cytometry. (**D**) Detection of IFN-γ production by ELISA after tumor-reactive CD8^+^ T cells co-cultured with tumor cells for 6 h. **** *p* ≤ 0.0001. (**E**) Quantification of tumor-induced IFN-γ production, CD137 and CD107a expression of CD8^+^ T cells in PBMCs before co-cultured or obtained by day 5 and day 10 of co-culture with tumor cells. All data were representative of at least three independent experiments.

**Figure 6 cancers-15-00251-f006:**
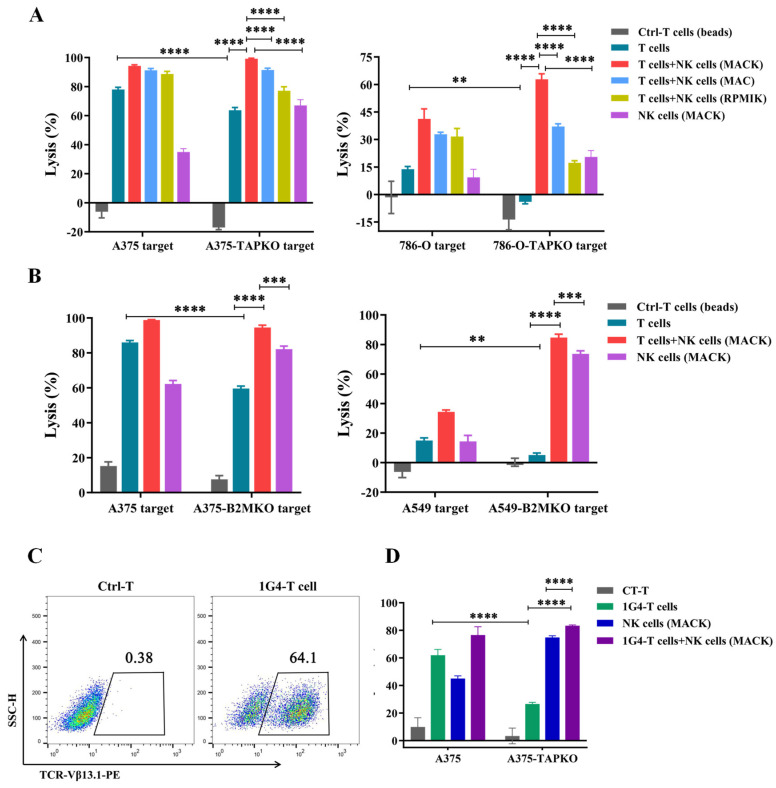
The killing efficiency of tumor-reactive T cells or NY-ESO-1-specific 1G4-T cells in combination with NK cells. (**A**) Calculated data were displayed to show the strength of NK cells cooperating with tumor-reactive T cells to eliminate A375 and 786-O cells, and corresponding cells with downregulated expression of HLA-I after knockout TAP gene (tumor-TAP KO). (**B**) The bar graph shows the results of combined cytotoxicity of NK cells and tumor-reactive T cells after knockout the B2M gene in A375 and A549 cells. (**C**) Flow cytometry analysis of 1G4 TCR expression on activated T cells by staining anti-PE-Vβ13.1. The control was T cell without transduced 1G4 TCR lentivirus. (**D**) Cytotoxicity of NY-ESO-1-specific 1G4-T cells in combination with NK cells against A375 cells and A375-TAP KO cells. Statistical analysis by two-way ANOVA with Tukey’s correction for multiple comparisons. ** *p* ≤ 0.01; *** *p* ≤ 0.001; **** *p* ≤ 0.0001. All data are representative of three independent experiments.

**Figure 7 cancers-15-00251-f007:**
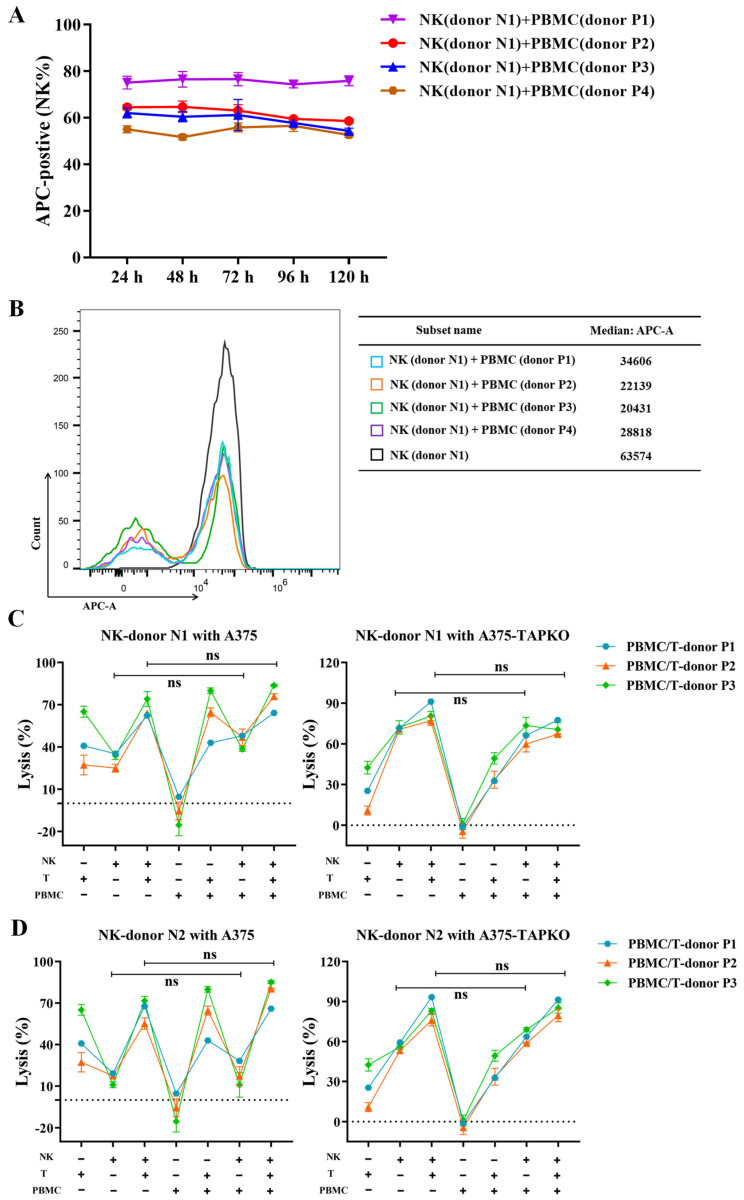
The effect of allogeneic PBMCs on the persistence and cytotoxicity of NK cells. (**A**) APC fluorescence-positive percentage was detected by flow cytometry every 24 h to track the persistence of NK cells that were labeled with the dye eFluor™ 670 after co-culture with different allogeneic PBMCs. (**B**) The labeled NK cells were co-cultured with PBMCs, represented as dividing populations. The mean fluorescence value of each group was calculated. The histogram presented the fifth day of co-culture, and the group of labeled NK cells cultured alone was represented as the initial labeling control. (**C**,**D**) The indicated combinations of NK cells from donor N1 (**C**) or donor N2 (**D**), tumor-reactive T cells and PBMCs were added to co-culture with A375 or A375-TAP KO. The lysis capability on target cells of each group was presented. Experiments were performed with tumor-reactive T cells and PBMCs from 3 different peripheral blood donors. Statistical analysis by two-way ANOVA with Tukey’s correction for multiple comparisons. ns represents not significant, corrected for multiple comparisons. All data are representative of three independent experiments.

## Data Availability

The data presented in this study are available in this article (and Appendix A).

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
