# Peer review of "Combination of Expanded Allogeneic NK Cells and T Cell-Based Immunotherapy Exert Enhanced Antitumor Effects"

_cancers, 2022, doi:10.3390/cancers15010251_

Round 1

Reviewer 1 Report

General

Strong selective pressure enforced on tumors by CD8 T cells following immune checkpoint blockade therapies or by adoptively transferred TILs, PBL-derived T cells or TCR-engineered T cells often results in tumor escape through deregulation or total loss of HLA-I presentation. The idea put forward and explored by in this manuscript is that this route for tumor escape can be blocked by the co-administration of NK cells, which are naturally geared to recognize and destroy HLA-I-low/negative cells.

This idea is not new. In fact, one criticism concerning this manuscript is that the authors do not put enough emphasis on accumulating evidence of total HLA-I loss in tumors following immunotherapy and on attempts to harness MK cells for counteracting relapse in these cases. A major hurdle to employing NK cells for this purpose stems from their refractoriness to ex-vivo propagation and expansion to the large amounts required for their therapeutic task. In this manuscript the authors describe a new protocol for preparing and expanding NK cells, which may offer a practical solution to this major hurdle.

The main tool exploited by the authors for achieving this goal was the HLA-I-negative lymphoblastoid cell line K562, which is widely employed as an artificial antigen-presenting cell (aAPC), mostly for ex-vivo propagation of human T cells. They stably transduced K562 cells with a battery of genes, encoding CD64, CD86, 4-1BBL and membrane-attached IL-21, all known for their ability to deliver important growth and costimulatory signals to human NK cells. After verifying proper cell surface expression of all gene products in the modified K562 cells (mK562) they went on to establish optimal growth conditions for PBMC-derived human NK cells following two weeks of co-culture, testing two different NK growth media. Indeed, they demonstrate high purity and improved fold expansion of NK cells grown under these conditions over NK cells grown in the absence of mK562 cells. They then monitored activation of the resulting NK cell cultures by different solid and hematological tumor cell lines, including cytotoxicity, and again showed superiority of cells grown in the presence vs. absence of mK562.

Subsequent RNA-seq analysis of NK cells revealed differences in the patterns of gene expression among cells propagated under the different culturing protocols. Interestingly, the upregulation of activatory receptors was linked to downregulation of inhibitory ones, and these correlated with the strength of the cytolytic activity.

Next, the authors performed a comprehensive bioinformatics analysis of genomic alterations in human cancers and identified TAP1/2 and B2m as key genes in the HLA-I processing and presentation pathway that are altered in a significant proportion of tumors. To explore the ability of the ex-vivo cultured NK cells to recognize and respond to cells with impaired HLA-I expression they knocked out these genes in a number of tumors. They then showed that the expanded NK cells selected according to their previous characterization exhibited enhanced cytolytic activity against the HLA-I-low tumor cells compared to the wild-type tumors.

To establish an experimental system enriched for tumor-reactive-T cells the authors used a special protocol, originally developed for enriching tumor-reactive autologous T cells by co-culturing PBLs and tumor organoids. They used these cells for examining whether the NK cells can enhance the anti-tumor T cell response. Results of these experiments were generally positive but differed among the tumor cells tested. Similar conclusion were also drawn when the antitumor T cells were transduced to express the anti-NY-ESO-1157-165/A2 TCR 1G4.

In the last experiment the authors ruled out an allogeneic response mounted against NK cells by allogeneic PBMCs at least during 5 days of co-culture, assessing allorejection of future universal ‘off-the-shelf’ NK cell products by the recipient immune system.

Overall evaluation

The approach explored in this study addresses a major challenge in adoptive cell therapy of cancer. The establishment of an efficient, safe and reproducible protocol for generating clinical off-the-shelf NK cell products would constitute an essential breakthrough in this technologically demanding field. However, there are several issues that need to be clarified, or corrected before this manuscript can be published. These are detailed in the list of comments below.

Comments

·       Meticulous editing and proofreading of the entire manuscript are required.

·       The authors are requested to re-examine their reference list. Several references (see below) are not entirely related to, or supportive of the relevant message in the text.

·       A growing number of studies report on the total loss of HLA-I by tumor cells, often resulting from loss of B2M. These papers discuss the role of NK cells under these circumstances and their potential use for blocking this escape route (see for example refs. (1–5)). The authors are requested to refer to such studies and their conclusions.

·       Upregulation of activatory receptors and downregulation of inhibitory ones, as shown in Fig. 3, raises a severe safety concern, especially in light of the considerable basal activity of the NK cells against HLA-I proficient tumor cells. The authors are requested to discuss this finding from the safety angle.

·       In line 467 the authors write: “…co-cultured tumor-reactive T cells with matched tumor cells…” but the exact match is not specified. The original protocol used for generating tumor-reactive T cells employed tumor organoids with autologous PBLs, producing genuine tumor-specific T cells. This is not the case here. Although the degree of HLA-I matching between the donor T cells tested and the allogeneic tumor is not clear (except perhaps for HLA-A2), the authors most likely generated alloreactive T cells against the foreign tumor cell lines. In the experimental system explored in this work this is not critical, but the authors are requested to refer to this notion.

·       In line 488, the authors claim that the tumor-reactive T cells they obtained using the above protocol were ‘neoantigen-reactive’ but there is no data to support this claim. What they observed was more likely, at least in large, alloreactivity against the mismatched HLA-I alleles expressed by the tumor cells. The authors are asked to refer to or rephrase this statement.

·       The results presented in Fig. 6 show considerable activity of NK cells against non-modified tumor cells, possibly raising a safety concern. Is this a natural anti-tumor response or an augmented response against ‘innocent’ cells owing to enhanced sensitivity of the NK cells?

·       Fig. 6A lacks the important control of NK cells only. Please add this control or provide a convincing explanation for its absence.

·       In the experiment shown in Fig. 7, how do the authors explain the apparent resistance of the NK cells tested to the allogeneic PBMCs?

·       Line 391: Please check if Ref. 30 is relevant here.

·       Line 579: Ref. 40 is a 2018 Cell paper and not as appears here. Please check and correct.

·       Line 581: Ref 42 in fact describes the opposite: TAP silencing induces, rather than abolishes, antitumor immunity, and this is certainly not supportive of the context of this citation.

Typos and minor comments

·       Line 19: Should be ‘are limited’.

·       Line 20: ‘Occurring’ is improper here.

·       Line 66: ‘10%-15%’ of what cells exactly?

·       Line 73: Correct ‘using’.

·       Line 77: Edit the entire sentence.

·       Line 112: Change ‘combinate’.

·       Line 256: Perhaps ‘Genes’?

·       Line 573: “…may occur immune incompetence…” ? Please correct.

References

1. Restifo, N. P., F. M. Marincola, Y. Kawakami, J. Taubenberger, J. R. Yannelli, and S. A. Rosenberg. 1996. Loss of functional beta 2-microglobulin in metastatic melanomas from five patients receiving immunotherapy. J. Natl. Cancer Inst. 88: 100–108.

2. Del Campo, A. B., J. A. Kyte, J. Carretero, S. Zinchencko, R. Méndez, G. González-Aseguinolaza, F. Ruiz-Cabello, S. Aamdal, G. Gaudernack, F. Garrido, and N. Aptsiauri. 2014. Immune escape of cancer cells with beta2-microglobulin loss over the course of metastatic melanoma. Int. J. cancer 134: 102–113.

3. Sade-Feldman, M., Y. J. Jiao, J. H. Chen, M. S. Rooney, M. Barzily-Rokni, J. P. Eliane, S. L. Bjorgaard, M. R. Hammond, H. Vitzthum, S. M. Blackmon, D. T. Frederick, M. Hazar-Rethinam, B. A. Nadres, E. E. Van Seventer, S. A. Shukla, K. Yizhak, J. P. Ray, D. Rosebrock, D. Livitz, V. Adalsteinsson, G. Getz, L. M. Duncan, B. Li, R. B. Corcoran, D. P. Lawrence, A. Stemmer-Rachamimov, G. M. Boland, D. A. Landau, K. T. Flaherty, R. J. Sullivan, and N. Hacohen. 2017. Resistance to checkpoint blockade therapy through inactivation of antigen presentation. Nat. Commun. 2017 81 8: 1–11.

4. Torrejon, D. Y., G. Abril-Rodriguez, A. S. Champhekar, J. Tsoi, K. M. Campbell, A. Kalbasi, G. Parisi, J. M. Zaretsky, A. Garcia-Diaz, C. Puig-Saus, G. Cheung-Lau, T. Wohlwender, P. Krystofinski, A. Vega-Crespo, C. M. Lee, P. Mascaro, C. S. Grasso, B. Berent-Maoz, B. Comin-Anduix, S. Hu-Lieskovan, and A. Ribas. 2020. Overcoming Genetically Based Resistance Mechanisms to PD-1 Blockade. Cancer Discov. 10: 1140–1157.

5. Paschen, A., I. Melero, and A. Ribas. 2022. Central Role of the Antigen-Presentation and Interferon-γ Pathways in Resistance to Immune Checkpoint Blockade. Annu. Rev. Cancer Biol. 6: 85–102.

Reviewer 2 Report

This study is mainly focused on optimization of preparation of allogenic activated NK cells for a cellular antitumor therapy, especially for cells with MHC class I expression defects. The  authors characterize the phenotype and functionality of expanded and activated human NK cells co-cultured  with specifically genetically engineered K562 cells. The second major topic is setting up a protocol for NK a T cell cellular therapy against MHC class I-deficient tumors. All experiments were performed in vitro using human cells.  This study is rather methodological, providing a lot of useful data, although not really priority biological findings. Anyway, it is of interest for researchers working in the field. MHC class I deficiency is a frequent mechanism by which tumor cells can escape from specific immune responses, which is often underestimated in the design of antitumor immunotherapeutic schemes.

Specific comments:

-          In Introduction section, citation 19 (Barbet G et al., 2021) id dedicated to the TAP dysfunction in dendritic, not tumor cells. Please add more citations on this topic.

-          In the Methods, detailed description of the K562 transduction is missing.

-          The  differences of the NK cells cellularity and phenotype that were co-cultured with transduced K562 cells vs. parental unmodified K562 cells has not been documented.

-          Do you know the MHC class I status of the tumor cell lines serving as the target cells in Fig. 2A?

-          Fig. 6, in some of the results, the NK cells treated only controls are not shown.

Round 2

Reviewer 1 Report

The authors have made an impressive and effective effort to respond to all comments and correct the MS accordingly.

Minor errors still must be corrected:

- 'Allogenic' appears several times and should be changed to 'allogeneic'.

- Lines 539-540, should be: no cases... 'were' observed.
